# Coatings of Cyclodextrin/Citric-Acid Biopolymer as Drug Delivery Systems: A Review

**DOI:** 10.3390/pharmaceutics15010296

**Published:** 2023-01-16

**Authors:** Karen Escobar, Karla A. Garrido-Miranda, Ruth Pulido, Nelson Naveas, Miguel Manso-Silván, Jacobo Hernandez-Montelongo

**Affiliations:** 1Departamento de Ciencias Matemáticas y Físicas, Universidad Católica de Temuco, Temuco 4813302, Chile; 2Agriaquaculture Nutritional Genomic Center, CGNA, Temuco 4780000, Chile; 3Departamento de Ingeniería Química y Procesos de Minerales, Universidad de Antofagasta, Antofagasta 1270300, Chile; 4Departamento de Física Aplicada and Instituto Universitario de Ciencia de Materiales Nicolás Cabrera, Universidad Autónoma de Madrid, 28049 Madrid, Spain; 5Instituto Universitario de Ciencias de Materiales Nicolás Cabrera, Universidad Autónoma de Madrid, 28049 Madrid, Spain

**Keywords:** cyclodextrins, cyclodextrin polymers, implant coating, drug delivery

## Abstract

In the early 2000s, a method for cross-linking cyclodextrins (CDs) with citric acid (CTR) was developed. This method was nontoxic, environmentally friendly, and inexpensive compared to the others previously proposed in the literature. Since then, the CD/CTR biopolymers have been widely used as a coating on implants and other materials for biomedical applications. The present review aims to cover the chemical properties of CDs, the synthesis routes of CD/CTR, and their applications as drug-delivery systems when coated on different substrates. Likewise, the molecules released and other pharmaceutical aspects involved are addressed. Moreover, the different methods of pretreatment applied on the substrates before the in situ polymerization of CD/CTR are also reviewed as a key element in the final functionality. This process is not trivial because it depends on the surface chemistry, geometry, and physical properties of the material to be coated. The biocompatibility of the polymer was also highlighted. Finally, the mechanisms of release generated in the CD/CTR coatings were analyzed, including the mathematical model of Korsmeyer–Peppas, which has been dominantly used to explain the release kinetics of drug-delivery systems based on these biopolymers. The flexibility of CD/CTR to host a wide variety of drugs, of the in situ polymerization to integrate with diverse implantable materials, and the controllable release kinetics provide a set of advantages, thereby ensuring a wide range of future uses.

## 1. Introduction

Medical implant surgery involves inherent risks, such as infections, local swelling and induration, inadequate healing, and others, which become more complicated in cases of immunosuppression caused by several diseases or conditions [1]. A very effective strategy to reduce these postoperative risks is to coat the implants with polymeric drug-delivery systems [2]. One of the most versatile drug-delivery systems is the cyclodextrin/citric-acid biopolymer (CD/CTR), which has been successfully used in the last years as a coating on different materials and implants for the local controlled release of several molecules. The synthesis of this biopolymer was developed by Martel et al. (2002) [3] and consists of the use of CTR, a polycarboxylic acid, as the cross-linking agent of CDs.

CDs are cyclic oligosaccharides with hydrophilic outer surfaces and a lipophilic central cavity, which allows the formation of reversible complexes with drugs and thus used as efficient delivery carriers [4]. CDs, as building blocks, are easily cross-linked with CTR, yielding a three-dimensional polymer network suitable for enhanced drug-delivery applications. Such volume structure presents extended interactions with drugs, prolonging residence time in the medium and/or increasing efficiency and specificity toward targeted sites [5].

The success of using CD/CTR on a wide range of surfaces is due to its cost-effectiveness and ease of in situ polymerization, which can be tuned by the reaction time at low temperatures. However, a pretreatment on the substrate surface is mandatory prior to performing the CD/CTR polymerization. This process is not trivial because it depends on the surface chemistry, geometry, and physical properties of the material to treat. Another important aspect of CD/CTR is the high capacity and diversity of molecules that can be loaded in its matrix, its biocompatibility [6,7], and biodegradability [8,9].

The most common mechanism of release of drug-delivery systems based on CD/CTR is the Fickian diffusion [10], which is a process in which the transport of the penetrant (water, PBS, or corporal fluids) is a diffusion process driven by the penetrant concentration gradient [11]. Usually, this phenomenon in CD/CTR systems is modeled by the semiempirical mathematical model of Korsmeyer–Peppas [12,13]. Although this tool is very useful for predicting the release kinetics before the release systems are realized, they must be used with criteria because they do not provide further insights into complex systems [14].

This review aims to cover the chemical properties of CDs, the syntheses of CD/CTR, and its applications as drug-delivery systems when coated on medical implants and other materials. The molecules released and other pharmaceutical aspects involved are discussed. Moreover, the different routes of the performed pretreatments on the substrates before the in situ polymerization of CD/CTR are also reviewed. The biocompatibility of the polymer was also highlighted. Finally, the mechanisms of release generated in the CD/CTR coatings were analyzed, including the Korsmeyer–Peppas model, which has been the most successfully semiempirical mathematical model to explain the release kinetics of drug-delivery systems based on these biopolymers.

## 2. Cyclodextrins

CDs are low molecular-weight cyclic oligosaccharides synthesized from the enzymatic degradation of starch. There are three main families of native cyclodextrins comprising six to eight (α=6,β=7,γ=8) D-glucopyranose units linked through glycosidic bonds between carbon 1 and 4 by a covalent bond [15]. Cyclodextrins have the advantages of being nontoxic, soluble in water, easily modifiable, and highly bioavailable. In addition, they are accessible and very useful in the industry [16,17]. The main physical properties of native CDs are presented in Table 1.

CDs are characterized by having a hydrophilic outer surface and a hydrophobic cavity (Figure 1) that allows them to form a rigid, cone-shaped cavity, where it can accept host molecules [18]. In this way, they form inclusion complexes by noncovalent interactions, and without complex chemical reactions [19]. Because each host molecule is individually surrounded by a CD, advantageous qualities can be provided in the chemical composition and in the physical properties of the host [4], such as stabilization of substances sensitive to light or to oxygen [20], modification of the chemical reactivity of host molecules [21], fixation of highly volatile substances, improvement of the solubility of substances [22], and protection against the degradation of substances by microorganisms [23], among others.

As α-CD presents the smaller internal cavity, its structure is the most rigid and stable of native CDs [24]. This makes it resistant to enzymatic hydrolysis, so it has several applications in the food industry. Unlike to γ-CD, which has a wider cavity, this is the most flexible and less stable CD [25]. In fact, in some conditions can collapse and not maintain its toroidal shape. From a pharmaceutical point of view, β-CD presents the best-balanced characteristics to obtain the higher inclusion-complexing capacity. For example, the values of inclusion-complex formation constants for α-CD, β-CD and γ-CD with salicylic acid is 11, 65, and 13 M${}^{-1}$, respectively. When the complex is formed with ibuprofen, this tendency is sharper: 55, 2600, and 59 M${}^{-1}$ for α-CD, β-CD and γ-CD, respectively [25].

On the other hand, native CDs present a variable number of hydroxyl groups capable of chemically reacting to incorporate different functional groups, which makes the formation of a great variety of derivatives possible [26]. The substituents can advantageously contribute to the inclusion of some specific host molecules and improve the ability to solubilize and stabilize the molecules in comparison with native CDs [27]. However, derivative CDs tend to be significantly more expansive than native CDs. Some examples of derivative **β**-CDs are the (2- hydroxypropyl)-**β**-cyclodextrin (HP-**β**-CD), random methylated **β**-cyclodextrin (RM-**β**-CD) and sulfobutyl ether-**β**-cyclodextrin (SBE-**β**-CD) (Table 2) [28].

CDs and their derivatives can be used as controlled drug release systems due to their characteristic cavity and their ability to form reversible complexes with drugs (Figure 2). In this way, the host molecule is slowly released from the cavity without altering the physical, chemical and biological properties of the drugs. In addition, CDs prolong the residence time of the molecules in the diffusion medium and increase the efficiency and specificity toward the target tissues [8,29]. Being considered biologically inert, CDs have been applied as pharmaceutical excipients for numerous drug formulations. Cyclodextrins **α**, **β**, **γ** and the derivative HP-**β**-CD are currently recognized as safe by the FDA, and by the European Pharmacopoeia [28].

## 3. Cyclodextrin/Citric Acid Biopolymer

In some cases, monomer units of CDs cannot form inclusion complexes with certain hydrophilic or high molecular mass drugs. Therefore, to increase this interaction, CDs and their derivatives have been used as building blocks for the development of a wide variety of networks, and polymer assemblies [18]. The main strategies applied to develop polymers of CDs can be classified into three large groups: (i) grafting of CDs to preformed polymers, (ii) polymerization with acrylic groups, and (iii) cross-linking with bi- or multifunctional agents, such as epichlorohydrin, biepoxides, and diisocyanates [30]. In the latter, the resulting polymer consists of a three-dimensional network that is suitable for drug-delivery applications. Figure 3 shows some types of polymers containing CDs. These CDs polymers are usually employed for drug-delivery applications; however, when they include molecules as quaternized amines they also present antibacterial effect [17]. The use of epichlorohydrin as a cross-linking agent of CDs has been the most widely used method. However, it has the drawbacks of taking place in concentrated basic media, in addition to the cross-linking agent being a toxic and carcinogenic reagent. As a result, the polymers obtained according to this route are not compatible with pharmaceutical or food uses [31]. Concerns about sustainability and human and environmental toxicity are driving the application of green chemistry principles to polymer synthesis. In this context, notable efforts are being made to select safe cross-linking agents and solvents [32]. In that sense, CTR is an effective multifunctional monomer for diverse syntheses; it is a nontoxic cross-linking agent, which is a natural organic acid with a multicarboxylic structure [33]. Moreover, CTR is very versatile because, under the defined conditions, it leads to soluble and insoluble polymers [31]. It plays roles in coforming ester cross-linking to improve compatibility coordination, balance the hydrophobicity of the polymer network, and provide other hydrogen bonding and binding sites for biocompatibility [34].

The origin of CD/CTR dates back to 1988 when Welch reported that the 1,2,3,4-butanetetracarboxylic acid (BTCA), a polycarboxylic acid (PCA), was able to provide effective cross-linking for cotton cellulose for a durable press finishing [35]. Later, in 1996, Yang and Wang studied the mechanism of this reaction and showed that the esterification between PCAs and cellulose occurred with the intermediate of a five- or six-membered cyclic anhydrides [36]. Then, in 2002, Martel et al. developed a cross-linking method that, compared to the others previously proposed in the literature, exclusively used nontoxic and environmentally friendly reagents [37]. This method consists of crosslinking CDs with biocompatible PCAs, like CTR, in the presence of phosphorus catalysts. The reaction begins with the dehydration of the PCAs, which generates a cyclic anhydride intermediate that easily reacts with the hydroxyl groups of the CDs through an esterification reaction. Then, two of the remaining carboxylic groups of the reacted PCAs can form a second anhydride that also reacts with another CD molecule. This reaction was made possible when PCAs contained at least three neighbouring carboxylic groups, like CTR [37]. In this reaction, sodium hypophosphite monohydrate ($\mathrm{Na}H_{2}{\mathrm{PO}}_{2}$·$H_{2}O$) is used as catalyst. Figure 4 shows the cross-linking mechanism between CTR and CD [10]. CD can be one of the three native CD forms or HPβ-CD [31,37].

To incorporate the CD/CTR on implant surfaces or other materials, the copolymer is covalently grafted onto the medical device (for example, on cellulose surfaces), or fixed by physical interactions around fibres (for example, based on polyester or polypropylene) or is mechanically locked within biomaterial pores (for example, within porous polyvinylidene difluoride membranes) [38]. On the other hand, the performance of the polymerization on a sample depends on the curing temperature, the reaction time, and the CD/CTR ratio. For example, the solubility of the polymer decreases with increasing temperature and reaction time, and the swellability decreases with an increasing temperature and a decreasing CD/CTR molar ratio [31,37].

In the case of the polymerization yield, it can be obtained from the mass increase or by determining the amount of carboxylic groups present in the polymer [29,37]. In this way, the coating process can be optimized to achieve maximum polymerization performance at a given temperature and time [39].

## 4. Coatings of Cyclodextrin/Citric Acid Biopolymer as Drug-Delivery Systems

The properties of a material are often determined by its surface rather than its bulk composition. In fact, surfaces are technologically significant in several areas, such as catalysis, biomedicine, environmental remediation, etc., because they play a critical role in the performance and success of materials’ applications [40]. In recent years, the functionalization or coating of surfaces has become an essential topic in materials science and engineering. Here, the coating aims at improving the material’s performance or making it more suitable for a specific application [41].

Several studies have reported applications of CD-based coatings both in biomedical [42,43,44,45,46] and nonbiomedical applications [16,37,39,47,48,49,50]. On the one hand, in the context of nonbiomedical applications, CD-based coatings have been used to prevent corrosion problems and to remediate pollutants or dyes. On the other hand, for biomedical applications, the importance of CD-based surface coatings has increased, in terms of both biocompatibility and drug delivery [51,52,53], mainly intended as molecular delivery systems, such as the delivery of antibiotics [42,43] and drugs [44,45,46], among others.

The CD/CTR biopolymers have been widely used as drug-delivery systems when they are incorporated as coatings on different medical implants and other materials. The mechanisms involved in loading molecules consist of an inclusion of the drugs inside the CDs cavity, as well as less specific interactions with the cross-linked polymer network and its carboxylic groups available for supplementary interactions with the drug through hydrogen bonds [44]. Table 3, Table 4 and Table 5 present the most highlighted works performed during the last two decades, regarding the applications of coatings of cyclodextrin/citric-acid biopolymers as drug-delivery systems.

The most used CD/CTR polymer has been the β-CD/CTR due to the high capacity of loading and releasing diverse drugs of β-CD when cross-linked. In addition, β-CD is an accessible low-cost molecule [74]. Other CD/CTR polymers that have been tested include native and derivative CDs, such as α-CD/CTR, γ-CD/CTR, HP-β-CD/CTR, and Me-β-CD/CTR. The most frequent technique is to add β-CD/CTR to the surface of the substrate as a single coating by an in situ polymerization. However, due to the anionic characteristic of β-CD/CTR, it has been also used in layer-by-layer assemblies (LbL) in combination with cationic polymers [8,9,62,70]. Another frequently used CD/CTR polymer is the HP-β-CD/CTR, which due to its extra hydroxypropyl groups of HP-β-CD allows the inclusion of specific guest molecules, and ameliorates the ability to solubilize and stabilize drugs [66]. This polymer has been used as a single coating and combined with chitosan to produce nanofibers by electrospinning [71]. Nevertheless, HP-β-CD is a more expensive CD in comparison with its native version.

The fan of coated medical implants and materials with CD/CTR polymers is diverse. Among implants are several plastic devices like PVDF membranes for periodontology [54,55], polyester vascular prostheses [56], polyamide inguinal meshes [57], polypropylene meshes for the treatment of hernias [59], poly(2-hydroxyethyl methacrylate) contact lenses [32], PET textiles [9,61,62,70], meshes [44], and vascular prostheses [68], PLLA parietal reinforcement [66], and PTT textiles [72]. Organic and inorganic materials have been used as substrates for the CD/CTR coatings, such as cellulose papers [60,63,65], hydroxyapatite [58], porous silicon [10,67,69,73], and zeolite [34]. Metal medical devices include titanium disks [8], CoCr [64], and NiTiNOL stents [71].

The thickness of this type of coating is in the order of microns. This is going to depend on the CD/CTR polymer, substrate, and coating technique. Moreover, the homogeneity of the coating is going to be strongly influenced by the geometry of the substrate. For example, 2.5 μm-thickness of a homogeneous β-CD/CTR coating was achieved on nanoporous silicon (nPSi) and macroporous silicon (mPSi) (Figure 5(A1–A4)) [10]. In a contrasting way, a coating of HP-β-CD/CTR applied to PET visceral mesh was heterogeneous with polymer structures of tens of microns added to the mesh fibres (Figure 5(B1,B2)) [44].

On the other hand, when the LbL technique is used to assemble a CD/CTR polymer, its thickness could be controlled according to the number of bilayers. Figure 5(C1–C4) shows the increase of thickness of the chitosan-β-CD/CTR assembling on titanium disks: 5, 10, and 15 bilayers were 6, 8, and 10 μm, respectively [8]. However, it should be mentioned that the most recent work of Jahangard et al. (2022) reported a β-CD/CTR layer, on the order of nanometers, for the functionalization of nanozeolite [34].

The principal application of the developed systems has been the prevention of preoperative and postoperative risks in implant surgeries, mainly focused on minimizing the risk of bacterial infections. Thereby, the most used drugs in the systems have been antibiotics, with ciprofloxacin being the most common. Other antibiotics have been also used, depending on the system (CD/CTR-implant) and application, even alternative antimicrobial agents such as the dye methylene blue [62] and the metal ions Zn${}^{2+}$ and Cu${}^{2+}$ [72].

As systems evolved, new prominent medical applications were explored like ocular treatments [10,32], arterial wall healing [64], and treatments for restenosis [71] and atherosclerosis [73]. Regarding veterinary applications, Hernandez–Montelongo et al. (2014) [67] proposed a system to improve the effect of ingested antimicrobials in salmon aquaculture. In the case of industrial uses, Lavoine et al. (2014) [63] explored a strategy to prolong food shelf life.

In general, all the works reported that coated samples with CD/CTR polymers showed a higher capacity to load drugs than uncoated samples. In the same sense, the treated substrates displayed more prolonged and sustained release than the untreated substrates. For example, Lepretre et al. (2007) [55] reported that PVDF membranes for periodontology coated with various CD/CTR polymers released considerably higher amounts of antibiotic chlorhexidine than raw membranes. Untreated samples released only 1 mg/g in a few hours, whereas membranes grafted with β-CD/CTR, γ-CD/CTR and HP-β-CD/CTR released up to 12 mg/g of drug within 55, 65, and 80 days, respectively.

As most of the CD/CTR-implant systems were loaded with antibiotics to prevent bacterial infections, they were mainly tested with bacteria *S. aureus*, *S. epidermidis* (both Gram-positive) and *E. coli* (Gram-negative). Other tested bacterial strains were *Enteroccocus* sp., *B. subtilis*, *F. nucleatum*, *P. melaninogenica*, *A. actinomycetemcomitans* and *P. gingivalis*. For example, Figure 6A shows the antibacterial property of a PET textile coated with an LbL assembling of a cationic and anionic β-CD/CTR polymers, this system released the antimicrobial triclosan for 24 h [70], and coated textiles displayed the double of inhibition zone than uncoated samples.

In all reviewed works, the CD/CTR-implant systems showed stronger antibacterial activity than materials without the CD/CTR coating. In the cases in which other biomedical applications were explored, interesting results were also obtained. For example, Guzman–Oyarzo et al. (2022) [73] reported that the controlled release of polyphenols from coated microparticles of nanoporous silicon with β-CD/CTR displayed higher antiangiogenic activity than polyphenols just in solution to treat HUVEC cells. Figure 6B shows the significant reduction in the formation of tubules in the three concentrations when caffeic acid, a polyphenol, was microencapsulated. Authors also reported that coated microparticles presented a more time-controlled antioxidant effect than uncoated samples.

The CD/CTR are biopolymers, which means that they are biodegradable and biocompatible. Regarding biodegradability, a study about it is reported by Vermert et al. (2017) [66]: samples of PLLA coated with HP-β-CD/CTR were immersed in PBS at 37 °C under constant stirring. The virgin PLLA did not reveal any significant weight loss after 180 days, whereas the functionalized PLLA (33%-wt) showed a weight loss of around 30% after 120 days and then stabilization until the end of the experiment. As a consequence, the observed weight loss was therefore attributed to the degradation of CD/CTR polymer coating by hydrolysis in PBS within four months. On the other hand, the biocompatibility of CD/CTR has been studied by using a wide range of cell strains, depending on the application, such as human embryonic lung cells (L132) [9,54], human epithelial embryonic cells [62], human pulmonary microvascular endothelial cells (HPMEC) [56,64], fibroblast cells (NIH3T3) [44], and human umbilical vein endothelial cells (HUVEC) [69,73].

Regarding biological assays using animals, there is a work reported in the literature by Jean-Baptiste et al. (2012) [75]. The authors coated polyester vascular prostheses (PVPs) with HP-β-CD/CTR polymer designed to provide an in situ reservoir for the sustained delivery of antibiotics (rifampin, vancomycin hydrochloride, and ciprofloxacine). Their results showed that coated samples presented excellent biocompatibility, healing, and degradation properties in vitro and in albino Swiss mice (*Mus musculus*) in an animal model. The antimicrobial activity of samples was shown against *S. aureus* and *E. coli.* strains.

It is important to mention that CD/CTR polymers have not only been used as drug-delivery systems, but other interesting applications have also been studied, such as the removal of pollutants in water of heavy metal ions [76,77], dyes [78,79], organic compounds [78,79,80], and other things. Most recently, CD/CTR has also been utilized for host–guest recognition in sensors of herbicides [81] and antibiotics [82].

## 5. Pretreatments on the Substrates

As shown in Table 3, Table 4 and Table 5, different surfaces or substrates have been coated with CD/CTR-based biopolymers to generate controlled drug-release systems. These functionalized surfaces have been used in different applications, such as tissue regeneration [54] and bacterial infections [58], along with medical supplies and surgical materials [60,62], among other things. Here, preparing the surface prior to coating application is essential to determining the properties of the coating, particularly its roughness, adhesion, corrosion resistance and the ability to deliver the therapeutic substances. Then, depending on the substrate to be coated, different surface pretreatments and functionalizations have been carried out in order to achieve a successful coating with the desired properties.

Porous materials are one of the most interesting types of materials for drug-delivery applications. This is due to advanced properties, such as high surface area and high reactivity, among other things. In this sense, several reports have shown porous silicon (PSi) as a promising material for drug delivery. For example, nanoporous and macroporous PSi has been coated with CD/CTR-based coating via in situ polymerization. Herein, the surface has been previously treated by a chemical oxidation step by using H_2_O_2_ to mechanically stabilize the pore structure [10,67,69,73,83]. Besides, this chemical treatment of PSi, prior to CD/CTR-based coating, is an oxidation process involving different reactions; Si–H bonds can be transformed to Si–OH, Si–O–Si or –Oy-Si–Hx. Thus, the amalgam between the PSi substrates and CD/CTR was generated by hydrogen bonding and the anchoring effect of the porous matrix. Moreover, the CD/CTR coating adhesion on PSi can be enhanced with a chitosan grafting after the PSi chemical oxidation; positive −NH_3_ of chitosan interacts electrostatically with both negative SiOx groups of the oxidized PSi surface, and negative −COOH of CD/CTR biopolymer [67,69,73].

Other porous materials, such as hydroxyapatite and zeolite, have been functionalized with CD/CTR-based coatings without any substrate pretreatment [34,58]. A similar methodology has been reported for PDVF-based substrates [54,55]. In both cases, hydrogen bonding was the main chemical interaction between substrates and CD/CTR.

In the case of metallic samples, such as CoCr stents and Ti disks, an elaborated pretreatment was necessary to succesfully graft the CD/CTR biopolymer [8,64]. First, samples were mechanically polished, followed by chemical oxidations by using a “piranha solution”. Afterward, samples were functionalized with dopamine, which enabled further CD/CTR-based coatings.

On the other side, by use of LbL techniques consisting of alternating physisorption of oppositely charged polyelectrolytes [84], different coatings have been formed on the substrate based on PET textil [9,61,62,70]. Interestingly, raw material activation was necessary for CD-CTR multilayer-based materials on nonwoven PET textile substrates. Such activation ensured the adhesion of the multilayer coating on the textile. This process is called “pad-dry-cure”. There are four steps that mainly constitute this process: (i) impregnation of the PET textile into an aqueous solution containing CTR as a crosslinking agent, CD and a catalyst, (ii) roller squeezing, (iii) drying, and (iv) thermofixing (at variable temperature and time) [37]. This process aims at developing an ion exchange textile by forming a coating based on the previously explained polyesterification reaction (Figure 4).

The pad-dry-cure process has also been used for functionalized visceral mesh, and vascular prostheses based on PET substrates [44,56,68], as well as the coating of synthetic thermoplastic polymers such as polypropylene and polyamide, which have been used as artificial abdominal wall implants for the prolonged release of ciprofloxacin and to obtain inguinal meshes with improved antibiotic delivery properties, respectively [57,59]. This methodology has also been used to coat biocompatible synthetic polymeric substrates such as poly-L-lactic acid, which is capable of inducing subclinical inflammation in the host to stimulate collagen formation [66,85].

For the cases of natural textile-based substrates, such as cotton and paper, both direct [60,63] or indirect coating followed by the pad-dry-cure method has been used to form CD-based controlled drug-release systems. In this line, Tabary et al., (2014) proposed the use of a prefunctionalization step based on the oxidation of paper points [65]. The oxidation reaction was done in a nitric acid (HNO_3_) and phosphoric acid (H_3_PO_4_) mixture at room temperature for 12 to 48 h with intermediate stirring with a glass rod. Their results showed that the functionalized oxidized paper point allowed the control of the bioresorption rate of the device, which could be used as a reliable complementary periodontal therapy. In this context, it is important to mention that natural textile-based substrates are mainly composed of cellulose, which implies that the esterification reaction occurs at the -OH functional groups of cellulose (or CDs), resulting in the formation of ester functional groups not present in cellulose. Thus, the detection of these functional groups determines the success of the CD grafting reaction on cellulose-based materials [60].

## 6. Mechanisms of Release of CD/CTR-Based Biopolymers

In clinical practice, when an implant surgery is performed, systemic treatments with various drugs are used to avoid or reduce possible postoperative complications. The methods consist of preoperative, perioperative, or postoperative administration of antibiotics, antiseptics, or other suitable molecules. Drugs can be administered orally, intravenously, or irrigated at the implantation site [86]. The doses administered and the duration of treatment are intensified in high-risk patients, such as those undergoing reconstructive surgery and/or undergoing radiotherapy treatments [87]. Traditional drug administration treatments, such as the venous or oral route, have poor control over these substances in the blood plasma because the drug concentration level reached is not stable. At the beginning of the administration, the level reaches high values, and then falls rapidly. Consequently, this variation in the concentration of the drug makes traditional treatments ineffective [88]. On the other hand, drugs given systemically can cause unwanted effects. For example, traditionally administered antibiotics may have difficulty penetrating the biofilm of bacteria, so only subinhibitory concentrations are reached at the site of infection, which causes the development of bacterial resistance [89]. Other drugs, such as zafirlukast (ZFL), which is administered orally for capsular contracture in mammary implant surgeries, can produce side effects like potential serious liver poisoning. For this reason, the use of ZFL is still restricted to severe and recurrent contractures [90].

The use of CD/CTR coatings on the implant surface offers a solution to the difficulties involved in the traditional drug-delivery system in surgeries. The release of the drug from the coated implant allows it to achieve a release in situ, that is, in the immediate environment of the implant. Thus, the side effects caused by the drugs can be reduced [59]. Moreover, the controlled delivery system based on CD/CTR coatings allows the drug concentration to be kept as close to the effective level of the treatment, which is below the toxic level and above the minimum of efficacy. In this way, the administration of repeated doses of drugs can be avoided [88].

On the other hand, in the drug-delivery systems based on polymers, the release can be controlled by diffusion, swelling, erosion, and/or external stimulations [91]. To develop and understand the drug-release mechanisms, mathematical models are very useful. In particular, semiempirical mathematical models are easy to use, and the established empirical rules help to explain transport mechanisms. In most cases, drug-diffusional mass transport is the predominant step in the control of drug release, whereas in others, it is present in combination with polymer swelling or polymer erosion [92]. In that sense, the most successfully semiempirical model to explain drug-delivery systems based on CD/CTR biopolymers is the Korsemeyer–Peppas model.

The Korsmeyer–Peppas model is a simple relationship to describe drug release from a polymeric system by using an exponential-type equation. This model analyzes Fickian, and non-Fickian release mechanisms considering the geometric characteristics of the system [93]. It was developed by Korsmeyer et al. (1983) [12], and Ritger and Peppas (1987) [13].

The Korsmeyer–Peppas equation is
(1)$$\begin{array}{c} Q(t)/Q_{\infty }=k_{\mathrm{KP}}t^{n}, \end{array}$$
where Q(t) is the amount of drug released in a time *t* and $Q_{\infty }$ is the amount of drug released in an infinitely long time; $k_{\mathrm{KP}}$ (h${}^{-n}$) is the Korsmeyer–Peppas kinetic constant, which characterizes the drug-matrix system and is also considered the release velocity constant; and *n* is the exponent that indicates the drug-release mechanism.

The release mechanisms, according to the Korsemeyer–Peppas model, are shown in Table 6. These models can be categorized into five types [93,94,95,96]: (1) the quasi-Fickian model indicates that drug diffusion is the predominant phenomenon, but the matrix is partially swollen; (2) the Fickian mechanism (referred to as “case I”) means that diffusion is the main present phenomenon; (3) in anomalous transport, the velocity of solvent diffusion and the polymeric relaxation speed possess similar magnitudes; (4) the zero order (“case II”) is when the drug is released at a constant rate independent of concentration, and the velocity of solvent diffusion is less than the polymeric relaxation process; (5) in the super case transport, the velocity of solvent diffusion is much higher, causing an acceleration of solvent penetration. This causes tension and breakage of the polymer (erosion).

In this regard, when Korsmeyer–Peppas model was applied to the drug release profiles from porous silicon coated with β-CD/CTR, the calculated mechanism release was quasi-Fickian for ciprofloxacin, prednisolone, florfenicol, and caffeic acid [10,67,69]. However, in the case of pinocembrin, the mechanism release was an anomalous transport due to the strong interaction between this polyphenol and the polymer [69]. Figure 7 shows the polyphenols release profiles, caffeic acid and pinocembrin, from porous silicon coated with β-CD/CTR. On the other hand, results obtained for simvastatin release profile from nanofibers based on HP-β-CD/CTR coating stents suggested that the erosion (super case II transport) was the mechanism release [34].

Although the Korsmeyer–Peppas model is a useful and easy mathematical tool to implement, it may only be used for up to 60% of the released drug, and it has no provisions for an analysis of the remaining 40%. Thus, there is no phenomenological connection with this data [97]. For this reason, it is important to develop mathematical models to adequately study the release kinetics from polymer coatings to liquid media.

In that sense, recently, Hernandez–Montelongo et al. coated silicone breast implants with HP-β-CD/CTR loading the rose bengal (RB) dye. This experimental setup was used for developing a mathematical model that covers 100% of the released drug. The proposed model also contemplates a unidirectional recursive diffusion process which follows Fick’s second law while considering the convective phenomena from the polymer matrix to the PBS, where the drug is delivered, and the equilibrium of the polymer–liquid drug distribution. Results indicated that diffusion controls the delivery rate; however, as RB concentration increases, the equilibrium plays an increasing role, becoming the controlling mechanism for a longer time [14]. These sophisticated mathematical models also allow for determining the drug-concentration profile in the polymer matrix (Figure 8).

## 7. Concluding Remarks and Future Perspectives

Postoperative risks of medical implant surgeries, such as bacterial infections, sharp pain, inadequate healing, and other risks, can be reduced by local drug release. In that sense, because CD/CTR are biodegradable, biocompatible, economically feasible, and ecofriendly, they have been successfully used as coatings on implants, and other materials, for drug-delivery applications. The fan of utilized substrates is very wide, which includes plastic implants, porous ceramics and semiconductors, cellulose papers, and metallic devices. Even though the CD/CTR coatings have been obtained in the order of microns on macrostructures, recent works are being focused on the synthesis of nanolayers and the coating of nanomaterials.

Traditionally, the main application of the developed drug-delivery systems, based on CD/CTR biopolymers, was to reduce bacterial infections. Therefore, antibiotics have been the most common drug for loading and releasing in these systems. Current works are centred on alternative molecules like metal ions. As systems evolved, new prominent medical applications were explored, like treatments for restenosis, atherosclerosis, and arterial wall healing. In that sense, the perspective is that more complex biomedical applications are going to be studied. Although the CD/CTR systems are very versatile and can be used as drug delivery for a wide fan of drugs and molecules, it is recommended that one perform experimental tests for loading and releasing new drugs.

On the other hand, diffusion is the principal predominant step in the control of release from drug delivery based on CD/CTR systems; however, other phenomena can be present, such as swelling or erosion. In that sense, the semiempirical model of Korsmeyer–Peppas has been the easiest and most useful mathematical implemented tool for studying the release kinetics of drugs. Nevertheless, this model can only be used to take into account up to 60% of the cumulative drug release. In that sense, to glean deeper insight into the release kinetics, mathematical models for the specific systems are starting to be developed.

In general, this review presented that coatings of CD/CTR biopolymers have been thoroughly proven as excellent drug-delivery systems; consequently, in vivo experiments should be also considered in further studies. In addition, these systems have significant commercial potential to improve the quality of medical implant surgeries.

## Figures and Tables

**Figure 1 pharmaceutics-15-00296-f001:**
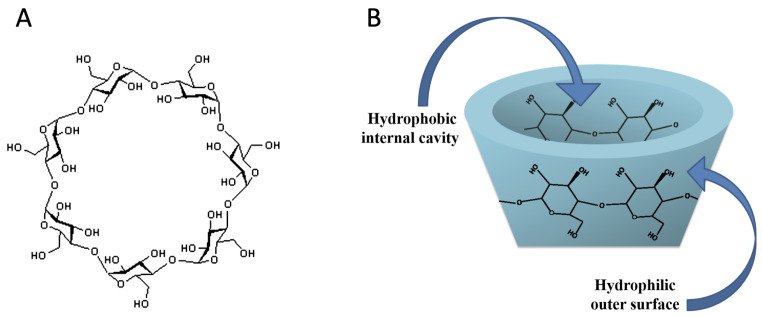
**β**-CD molecule: (**A**) chemical structure, and (**B**) toroidal shape.

**Figure 2 pharmaceutics-15-00296-f002:**
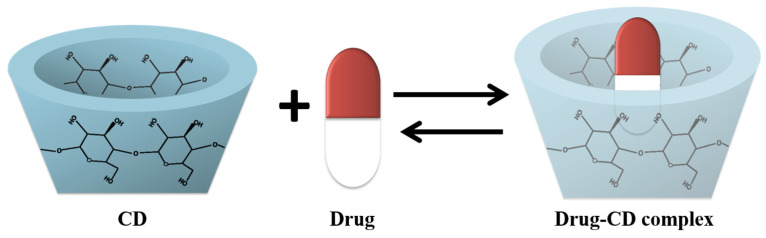
Dynamic equilibrium for drug-cyclodextrin complex.

**Figure 3 pharmaceutics-15-00296-f003:**
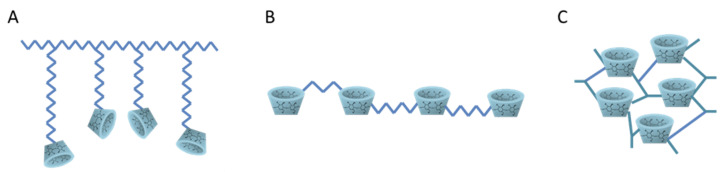
Types of polymers containing cyclodextrins: (**A**) pendent, (**B**) linear, and (**C**) cross-linked.

**Figure 4 pharmaceutics-15-00296-f004:**
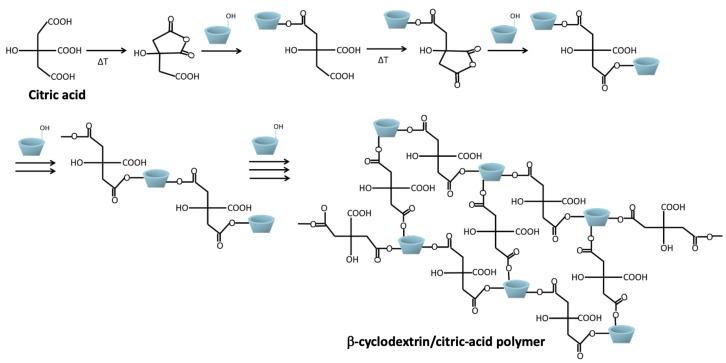
Polyesterification reaction between citric acid and β-cyclodextrin for the formation of the cross-linked polymer.

**Figure 5 pharmaceutics-15-00296-f005:**
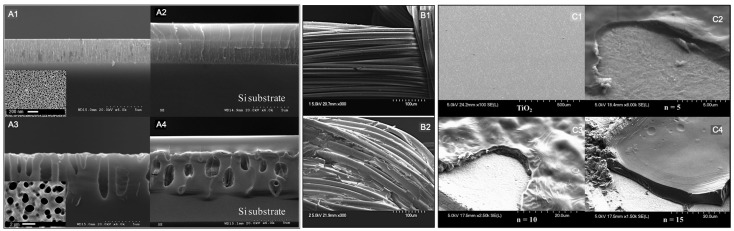
SEM images of: (**A1**) nPSi, (**A2**) nPSi-β-CD/CTR, (**A3**) mPSi, (**A4**) mPSi-β-CD/CTR; (**B1**) PET visceral mesh, (**B2**) PET-HP-β-CD/CTR, (**C1**) Ti, (**C2**) Ti-LbL of 5 bilayers, (**C3**)Ti-LbL of 10 bilayers, and (**C4**) Ti-LbL of 15 bilayers. Bilayers are chitosan and β-CD/CTR. Adapted from [8,10,44], with permission from Elsevier and ACS.

**Figure 6 pharmaceutics-15-00296-f006:**
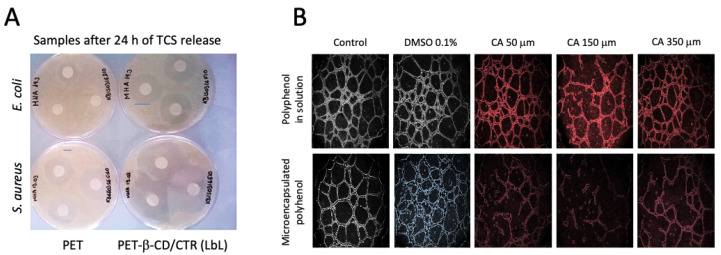
(**A**) Representative images of Kirby–Bauer test on TCS impregnated PET and PET-β-CD/CTR (LbL) samples against *S. aureus* and *E. coli* with inhibition zone after 24 h of TCS release in PBS at 37 °C. (**B**) Effect of treatment with caffeic acid in solution and loaded in the nPSi-β-CD/CTR microparticles on the ability of HUVECs to form tubular structures in Matrigel. Adapted from [70,73], with permission from Elsevier and MDPI, respectively.

**Figure 7 pharmaceutics-15-00296-f007:**
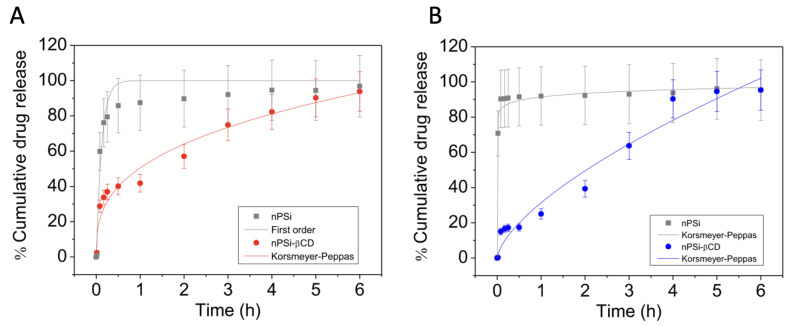
Polyphenols release profiles from porous silicon (control) and porous silicon coated with β-CD/CTR in phosphate-buffered saline (PBS) at 37 °C. (**A**) Caffeic acid, and (**B**) pinocembrin. Adapted from [69], with permission from MDPI.

**Figure 8 pharmaceutics-15-00296-f008:**
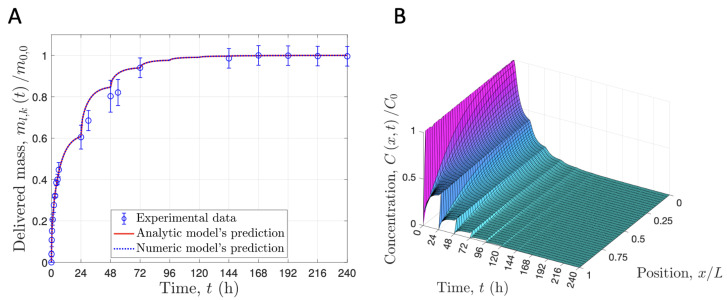
Recursive drug delivery. (**A**) Comparison of the experimental data and prediction for the delivered rose bengal. (**B**) Prediction of the drug concentration profile in the HP-β-CD/CTR matrix. Adapted from [14], with permission from MDPI.

**Table 1 pharmaceutics-15-00296-t001:** Characteristics of native cyclodextrins.

Property	α-CD	β-CD	γ-CD
D-glucopyranose units	6	7	8
Molecular weight (g/mol)	972	1135	1297
Solubility in water at 25 °C (% *w*/*v*)	14.5	1.85	23.2
Outer diameter (Å)	14.6	15.4	17.5
Cavity diameter (Å)	4.7–5.3	6.0–6.5	7.5–8.3
Height (Å)	7.9	7.9	7.9

**Table 2 pharmaceutics-15-00296-t002:** Characteristics of some derivative **β**-cyclodextrins.

Property	β-CD	HP-β-CD	RM-β-CD	SBE-β-CD
Number of substituted units	0	0.65	1.8	0.9
Molecular weight (g/mol)	1135	1400	1312	2163
Solubility in water at 25 °C (% *w*/*v*)	1.85	>600	>500	>500

**Table 3 pharmaceutics-15-00296-t003:** Applications of coatings of cyclodextrin/citric-acid biopolymers as drug-delivery systems (Part 1).

Coating	Substrate	Drug	Application	Key Results	Ref.
β-CD/CTR	Polyvinylidene difluoride (PVDF) membranes for periodontology	Doxycyclin (DOX) and chlorhexidine (CHX), antimicrobial agents	A membrane for guided tissue regeneration applicable in periodontology	Grafted membranes, delivered DOX and CHX in larger quantities within 24 h and 10 days, respectively, in comparison with raw membranes than delivered low amounts of both molecules within the first few hours of tests; treated membranes showed biocompatibility to L132 cells	[54], 2006
β-CD/CTR, HP-β-CD/CTR, γ-CD/CTR and HP-γ-CD/CTR	Polyvinylidene difluoride (PVDF) membranes for periodontology	Chlorhexidine diacetate (CHX), an antiseptic	To cure periodontal lesions	Grafted membranes released CHX during 60–80 days, more than tenfold of raw membranes	[55], 2007
HP-β-CD/CTR	Polyester vascular prostheses	Ciprofloxacin, vancomycin, rifampicin; antibiotics	To minimize the risk of infection of arteries replacement surgeries	Higher amounts of the antibiotics were absorbed in the coated prostheses compared to the pristine ones, which was reflected in the microbiological tests (*S. aureus*, *E. coli* and *Enteroccocus* sp.); coated samples showed proliferation of HPMEC cells	[56], 2008
β-CD/CTR, HP-β-CD/CTR, γ-CD/CTR and HP-γ-CD/CTR	Polyamide inguinal meshes	Ciprofloxacin (CFX), an antibiotic	To avoid bacterial contamination of inguinal wounds	Meshes grafted with HP-γ-CD/CTR presented a 10-fold CFX absorption than raw samples, and also showed a longer antimicrobial effect (*S. aureus*, *S. epidermidis*, *E. coli*); fibroblastic cells proliferated on grafted samples	[57], 2008
HP-β-CD/CTR	Microporous hydro-xyapatite (HA)	Ciprofloxacin (CFX) and vancomycin (VCM), antibiotics	To prevent preoperative infection in bone-graft surgeries	Functionalized HA showed prolonged antibiotics release and higher antibacterial activity against *S. aureus* than pristine HA; treated samples presented cytocompatibility to osteoblasts	[58], 2009
HP-γ-CD/CTR	Polypropylene (PP) meshes for the treatment of hernias	Ciprofloxacin (CFX), an antibiotic	To prevent abdominal postoperative wound infections	Microbiological tests using *S. aureus*, *S. epidermidis*, *E. coli* confirmed the higher sustained antibacterial activity of the coated meshes than uncoated samples; treated samples did not affect fibroblast proliferation	[59], 2011
β-CD/CTR	Three papers used for sterilization: noncoated paper, a 10% cotton-based medical bandage, and a medical crepe paper	Chlorhexidine digluconate (digCHX), an antiseptic agent	To provide bactericidal properties to cellulose-based materials of medical use	All functionalized papers maintained the release for periods up to 20 days	[60], 2013
α-CD/CTR, β-CD/CTR, γ-CD/CTR and HP-β-CD/CTR	Poly(2-hydroxyethyl methacrylate)-based contact lenses	Ethoxzolamide (ETOX), a carbonic anhydrase inhibitor	An ocular treatment for glaucoma	Ionic interactions between CD polymers and contact lenses sustained the ETOX release for several weeks	[32], 2013
LbL of chitosan and β-CD/CTR	Nonwoven polyethylene terephthalate (PET) textile	4-*tert*-butyl benzoic acid (TBBA), an antimicrobial agent	The release of antimicrobial agents and antiproliferative drugs in cancer therapies	TBBA release kinetics was controlled by the number of layers in the LbL system	[61], 2013

**Table 4 pharmaceutics-15-00296-t004:** Applications of coatings of cyclodextrin/citric-acid biopolymers as drug-delivery systems (Part 2).

Coating	Substrate	Drug	Application	Key Results	Ref.
LbL of chitosan and β-CD/CTR	Nonwoven polyethylene terephthalate (PET) textile	Methylene blue (MB), a cationic dye with antimicrobial properties	To design biomaterials to trap and release therapeutic molecules directly to targeted areas	LbL systems displayed sustained antibacterial effects against *S. epidermidis* of the textile through the MB prolonged release; samples were cytocompatible to human epithelial embryonic cells	[62], 2013
β-CD/CTR	Layers of nanoporous and macroporous silicon (nPSi and mPSi)	Ciprofloxacin (CFX), an antibiotic; and prednisolone (PDN), an antiinflammatory	Intraocular drug delivery system for postophthalmic surgery	Both functionalized samples controlled released therapy concentrations of CFX and PDN required for an adult human eye. Treated samples presented cytocompatibility to L132 cells	[10], 2014
A bilayer of β-CD/CTR and microfibrillated cellulose (MFC)	Food packaging paper	Carvacrol, an antibacterial molecule	A strategy to prolong food shelf-life	Treated paper sustained four times the drug than raw paper, and were antibacterial for 14 h against *B. subtilis*	[63], 2014
Me-β-CD/CTR	CoCr vascular stents	Paclitaxel (PTX), a highly hydrophobic anticancer agent	To obtain commercial stents that promote arterial wall healing	Coated stents held more PTX over time compared to the uncoated samples in human plasma, and were cytocompatible to HPMEC cells	[64], 2014
β-CD/CTR	Paper points (PP) for endodontic therapies	Chlorhexidine digluconate (digCHX), an antiseptic agent for periodontal therapies	A treatment of the periodontal pocket by preventing its recolonization by the subgingival microflora	Coated PP showed a prolonged release of digCHX in human plasma and sustained antibacterial activity against four periodontal pathogens: *F. nucleatum*, *P. melaninogenica*, *A. actinomycetemcomitans* and *P. gingivalis*	[65], 2014
HP-β-CD/CTR	Visceral mesh of (polyethylene terephthalate, PET fibres	Ropivacaine, an anaesthetic	To reduce postoperatory pain	The coated meshes impregnated in 10 mg/mL ropivacaine solution adsorbed up to 17.7 mg/g of drug, with a prolonged release of 100 min; coated samples loaded with ropivacaine showed cytocompatibility with NIH3T3 cells (fibroblasts)	[44], 2014
LbL of chitosan and β-CD/CTR	Titanium disks	Gentamicin, an antibiotic	To address perioperative infections	The amount of loaded drug was easily controlled by modulating the number of layers involved in the LbL system; coated disks exhibited microbial activity up to 6 days against *S. aureus*	[8], 2015
LbL of epichlorohydrin-glycidyltrimethyl-ammoniumchloride-β-CD and β-CD/CTR	Non-woven polyethylene terephthalate (PET) textile	4-*tert*-butyl benzoic acid (TBBA), an antimicrobial agent	To reduce the risk of infection in implantable PET biomaterials	Thermal cross-linking of the LbL system enhanced the stability and TBBA release kinetics, which was reflected in its high antibacterial effect against *S. aureus*, and *E. coli*; samples were noncytotoxic to L132 epithelial cells	[9], 2016
HP-β-CD/CTR	Poly-L-lactic acid (PLLA) parietal reinforcement	Ciprofloxacin (CFX), an antibiotic	To prevent bacterial infections in surgeries of hernias of the abdominal wall	The cytocompatibility with fibroblasts of meshes, and the antibacterial effect of CFX against *S. aureus* and *E. coli.*, were found to be dependent on the degree of functionalization	[66], 2017

**Table 5 pharmaceutics-15-00296-t005:** Applications of coatings of cyclodextrin/citric-acid biopolymers as drug-delivery systems (Part 3).

Coating	Substrate	Drug	Application	Key Results	Ref.
β-CD/CTR	Nanoporous microparticles (nPSi)	Florfenicol (FF), the most important antibiotic employed in aquaculture	To efficientize the medical effect of ingested antimicrobials in salmon aquacultures	Treated samples allowed a major control in the drug time release kinetics compared to raw samples, in both distilled water and simulated seawater	[67], 2018
Me-β-CD/CTR	Woven polyethylene terephthalate (PET) vascular prostheses	Ciprofloxacin (CFX), an antibiotic	To reduce the risk of synthetic vascular graft infection (SVGI), a postoperative infection	CFX release from virgin prostheses was faster than from functionalized prostheses	[68], 2019
β-CD/CTR	Nanoporous microparticles (nPSi)	Caffeic acid (CA) and pinocembrin (Pin), polyphenols	A safe alternative system for oral administration of polyphenols	Coated microparticles loaded higher amounts of both polyphenols, which also showed a better-controlled release than uncoated samples; treated microparticles presented cytocompatibility to HUVEC cells	[69], 2019
LbL of epichlorohydrin-glycidyltrimethyl-ammoniumchloride-β-CD and β-CD/CTR	Nonwoven polyethylene terephthalate (PET) textile	Triclosan (TCS), a broad spectrum antimicrobial agent	To reduce the risk of infection in implantable PET biomaterials	Treated textile loaded TCS four times than control and displayed a high and constant level over at least 28 days that pristine textile; treated samples showed intrinsic contact killing property and also extrinsic release killing against *S. aureus* and *E. coli*	[70], 2020
Nanofibers (NFs) of chitosan and HP-β-CD/CTR produced by electrospinning	Auto-expansible NiTiNOL stents	Simvastatin (SV), a lipid-lowering medication	To prevent restenosis	The extension of the release time of SV depended on the duration of electrospinning and on the presence of HP-β-CD/CTR in the NFs matrix	[71], 2020
β-CD/CTR	Polytrimethylene terephthalate (PTT) textiles	${\mathrm{Zn}}^{2+}$ and ${\mathrm{Cu}}^{2+}$ as antibacterial metal ions	To obtain an ecofriendly biobased antibacterial system for PTT fabrics	Coated textile showed an antibacterial effect to *S. aureus* and *E. coli*; ${\mathrm{Cu}}^{2+}$ displayed a stronger antibacterial ability	[72], 2021
β-CD/CTR	Nanoporous microparticles (nPSi)	Caffeic acid (CA) and pinocembrin (Pin), polyphenols	To improve the antiangiogenic and antioxidant activity of CA and Pin for the treatment of atherosclerosis	Coated microparticles showed higher antiangiogenic activity of CA and Pin than both in solution to treat HUVEC cells; in addition, coated microparticles presented a more time controlled antioxidant effect than uncoated samples	[73], 2022
β-CD/CTR	Nanozeolite	Ibuprofen (IB), a nonsteroidal antiinflammatory drug	An ecofriendly platform for IB delivery	β-CD/CTR-nanozeolite containing IB (30 wt%) showed the highest release at pH = 3.6 within the first 3–48 h of release time	[34], 2022

**Table 6 pharmaceutics-15-00296-t006:** Release mechanisms for different geometries according to the Korsmeyer–Peppas model.

Release Mechanisms	Geometry	Release Exponent *n*
Quasi-Fickian	Planar (thin films)	*n* < 0.5
	Cylinders	*n* < 0.45
	Spheres	*n* < 0.43
Fickian diffusion (case I)	Planar (thin films)	0.5
	Cylinders	0.45
	Spheres	0.43
Anomalous transport	Planar (thin films)	0.5 < *n* < 1
	Cylinders	0.45 < *n* < 1
	Spheres	0.43 < *n* < 1
Zero order (case II)	Planar (thin films)	1
	Cylinders	0.89
	Spheres	0.85
Super case II transport	Planar (thin films)	*n* > 1
	Cylinders	*n* > 0.89
	Spheres	*n* > 0.85

## Data Availability

The data that support the findings of this study are available from the corresponding author upon reasonable request.

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
