# Peer review of "Coatings of Cyclodextrin/Citric-Acid Biopolymer as Drug Delivery Systems: A Review"

_pharmaceutics, 2023, doi:10.3390/pharmaceutics15010296_

Round 1

Reviewer 1 Report

The manuscript is well written.  I recommend publication. 

The authors can improve the comment about the influence of type of polymer with cyclodextrins (Fig 3) on the antibacterial properties. 

Reviewer 2 Report

This review is a good one and good job of compiling this information. It would be more helpful for the researchers if you added a table including the cost-effectiveness of CTR and a table including the toxicity studies for this mixture.

Reviewer 3 Report

This is an interesting study and the authors have collected a unique dataset using cutting edge methodology. Overall the information presented represents valuable information regarding the feasibility of using CD/CTR applications as drug delivery systems when coated on different substrates. However, there is no clear aim and several areas in the manuscript deserve improvements.

Reviewer 4 Report

Hernandez-Montelongo, J. and coauthors presented the review dealing with polymers based on cyclodextrins and citric acid, as a linker. Review was well written and organized, however, some concerns are pointed out below.

1) authors should improve their keywords by changing those that already appear in the title of the article. this strategy can increase the publication's scope.

2) I strong suggest authors to rephrase “The synthesis of this biopolymer was developed by Martel et al. (2002) [3], and consists of the use of CTR, a polycarboxylic acid, as the crosslinking agent of CDs.” In the introduction section. The mentioned reference [number 3] deals with according to the published manuscript “The present article aimed to discuss the complexing properties of these CD-grafted fabrics toward some aromatic compounds dissolved in aqueous media.”. In page four authors correctly mention the achievements obtained and presented in reference [3].

3) most of the presented information in section 3 have been reported in many other reviews, I suggest authors to bring some additional information to improve this section.

4) authors indicate in section 3 that cyclodextrins are inexpensive. I must disagree (e.g., 100g can vary from 100 to 600 US$, depending on the cyclodextrin derivative, purity and so on). This may reflect directly on the number of pharmaceutical formulations containing cyclodextrins, nowadays.

5) page 2,lines 66-67: remove apolar, keeping the hydrophobic cavity.

6) page 3, Figure 2: equilibrium arrow must be used for reversible process.

7) page 3, line 93: authors should clearly indicate what they mean by “monomeric CDs”.

8) major concern: since the authors stated that CTR may be a better crosslinker than other examples, including epichlorohydrin. A section must be added highlighting the cytotoxicity and biocompatibility of CD/CTR polymers versus those using another crosslinker.

9) authors should improve section 6, by including other drug release examples, in order to improve this section.

Reviewer 5 Report

The paper introduces the CD/CTR delivery system in detail, which is very meaningful and can provide certain perspectives for researchers. However, I have a few small questions for the authors

1.     Although CD are classified as three types, they are not specifically differentiated in the subsequent sections.

2.     β-CD is always used for host-guest recognition, maybe you can talk more about β-CD/CTR.

3.     Can a rule be summarized to guide the types of drugs that can be delivered by CD/CTR system?

4.     What is the role of cyclodextrin and citric acid in CD/CTR system?

5.     There is no R in citric acid, why is citric acid short for CTR?

Round 2

Reviewer 4 Report

The authors significantly improved the quality of the main text of the review based on the comments and suggestions of all reviewers. I still recommend an inclusion on biocompatibility, cytotoxicity and so on, comparing this copolymer with others using different linkers. I understand that this will bring more strength to the review based on the authors' indications that citric acid is low toxic and may be more applicable as a linker.